# The Hydrolytic Peptides of Soybean Protein Induce Cell Cycle Arrest and Apoptosis on Human Oral Cancer Cell Line HSC-3

**DOI:** 10.3390/molecules27092839

**Published:** 2022-04-29

**Authors:** Cheng-Hong Hsieh, Tzu-Yuan Wang, Bo-Chen Tung, Hui-Ping Liu, Lien-Te Yeh, Kuo-Chiang Hsu

**Affiliations:** 1Department of Food Nutrition and Health Biotechnology, Asia University, 500 Lioufeng Rd., Wufeng, Taichung City 41354, Taiwan; hyweuan@asia.edu.tw; 2Department of Medical Research, China Medical University Hospital, China Medical University, No. 2, Yude Rd., North Dist., Taichung City 40433, Taiwan; 3Department of Nutrition, China Medical University, No. 100, Sec. 1, Jingmao Rd., Beitun Dist., Taichung City 40604, Taiwan; adam860711@gmail.com (B.-C.T.); sayanoao@gmail.com (H.-P.L.); 4Division of Endocrine and Metabolism, China Medical University Hospital, 2 Yude Rd., Taichung City 40447, Taiwan; yuan.w16@gmail.com; 5Department of Baking Technology and Management, National Kaohsiung University of Hospitality and Tourism, No. 1, Songhe Rd., Xiaogang Dist., Kaohsiung City 81271, Taiwan; yld@mail.nkuht.edu.tw

**Keywords:** soy protein isolate, anticancer peptides, oral cancer, antiproliferation, cell cycle, apoptosis

## Abstract

Protein hydrolysates from various sources, including tuna cooking juice, soy protein isolate, sodium caseinate, wheat gluten and skin gelatin from porcine, tilapia, halibut and milkfish were analyzed to screen their antiproliferative activities against the human oral squamous carcinoma cell line, HSC-3. The soy protein isolate was selected for further investigations based on its hydrolysates with bromelain (SB) and thermolysin (ST), showing the greatest inhibition of cell growth. The SB and ST hydrolysates showed antiproliferative activities up to 35.45–76.39% against HSC-3 cells at 72 h, and their IC_50_ values were 0.74 and 0.60 mg/mL, respectively. SB and ST induced cell cycle arrest in the S phase through a pathway independent of p21 and p27 protein expression. Further, ST induced the apoptosis of HSC-3 cells by downregulating expression of Bcl-2, PARP, caspase 3 and caspase 9, but an upregulating expression of p53 and cleaved caspase 3. Unlike ST, SB may induce necrosis on HSC-3 cells. Thus, soybean hydrolysates may be a good source for providing antiproliferative peptides against HSC-3, while SB and ST may have the potential to be developed as functional foods.

## 1. Introduction

Oral cancer is one of the most frequently diagnosed cancers worldwide and is a complex and long-term multifocal disease [1]. The prevalence of oral cancer is particularly high among men and is the sixth most common cancer worldwide [2]. In Taiwan, oral cancer is the fifth terminal cancer. Even with the advent of modern techniques, treatment with radiotherapy and chemotherapy still provides little improvement in the survival rate of oral cancer.

Other studies have demonstrated that natural products may possess antitumor or antiproliferative activities against cancer cells. For example, grape seed extract [3], red algal methanol or ethanol extracts [4,5,6], have been reported antiproliferative to oral cancer cells. Recently, proteins and peptides from various plant or animal proteins, especially milk, soybean or fish proteins, have been shown to have antitumor or antiproliferative activities. For example, lactoferrin and lactoferricin from bovine milk had an inhibitory effect on lung metastasis and angiogenesis in mice xenografted with murine melanoma, lymphoma or colon carcinoma [7]. Bowman–Birk protease inhibitor (BBI), a water-soluble protein isolated from legumes or monocotyledonous seeds, has been shown to be anticarcinogenic in vitro and in vivo [8]. Hydrophobic peptides from defatted soy protein hydrolyzed with thermos showed cytotoxicity on mouse monocyte and macrophage cell lines [9]. Soybean Kunitz trypsin inhibitor had been reported to suppress ovarian cancer cell invasion by blocking urokinase upregulation [10]. Lunasin, a novel chemopreventive peptide from soybean, has been found to suppress chemical carcinogen and viral oncogene-induced transformation of mammalian cells and inhibit skin carcinogens in mice [11,12]. In our previous study, the hydrophobic peptides isolated from dark tuna muscle by-products and protein hydrolysates from tuna cooking juice had a dose-dependent inhibition effect on human breast cancer cell line MCF-7 [13,14]. In addition, roe protein hydrolysates of giant grouper (*Epinephelus lanceolatus*) were reported to inhibit cell proliferation of oral cancer cells [15].

It is surprising, however, that there were only a few related research reports on antiproliferative and apoptosis of oral cancer cell lines induced by peptides obtained from food proteins. Therefore, the objective of this study is to investigate if there is an HSC-3 cell cycle arrest and apoptosis associated with the proteolytic products of soybean sources.

## 2. Materials and Methods

### 2.1. Materials and Reagents

Wheat gluten, soy protein isolate and sodium caseinate (from cow’s milk) were purchased from Gemfont Corporation (Taipei, Taiwan). Tilapia fish skins, the processing byproduct recovered from fresh skin off fillets, were supplied by Fortune Life Enterprise Co., Ltd. (Kaohsiung, Taiwan); the milkfish skins were donated by Simmy Seafood Co., Ltd. (Long An Province, Vietnam); the halibut and Atlantic salmon (Salmo salar) skins were supplied by Albion Fisheries Ltd. (Vancouver, BC, Canada). The tuna cooking juice was donated by a canned tuna processor in Chiayi County (Taiwan). Gelatin (from porcine skin) was purchased from Sigma-Aldrich, Inc. (St. Louis, MO, USA). Bromelain (from pineapple) was purchased in dry powder form from St Bio, Inc. (Taipei, Taiwan). Thermolysin (from *Bacillus thermoproteolyticus rokko*), pepsin (from porcine gastric mucosa) and Protease XXIII (from *Aspergillus melleus*) were obtained in dry powder form from Sigma-Aldrich, Inc. (St. Louis, MO, USA). Alcalase (from *Bacillus licheniformis*) and Flavourzyme (from *Aspergillus oryzae*) were purchased from Neova Technologies Inc. (Copenhagen, Denmark). Other chemicals and reagents used were analytical grade and commercially available.

### 2.2. Extraction of Gelatin from Fish Skins

The thawed fish skins were gently washed with running tap water, drained, and cut into pieces (about 5 × 10 cm). The fish skins were soaked in 0.2 M NaOH (1:10; *w*/*v*) and stirred in a cold room at 4 °C for 30 min. This procedure was repeated three times to remove noncollagenous proteins and pigments. The skins were washed with running tap water until the pH of the rinsing water was neutral. Afterward, the skins were soaked in 0.05 M acetic acid (1:10; *w*/*v*), stirred at room temperature for 3 h, and then washed with running tap water until the pH of the rinsing water was neutral. The gelatin of the swollen skins was extracted in distilled, deionized water (1:2; *w*/*v*) at 70 °C for 3 h. The oil and aqueous layers of the extract were separated by separatory funnels, and the extract was filtered through two layers of cheesecloth, lyophilized, and stored in a desiccator at room temperature until use.

### 2.3. Preparation of Protein Hydrolysates

Wheat gluten (W), soy protein isolate (S) and sodium caseinate (C) hydrolysates were prepared using thermolysin (T) (E/S (enzyme/substrate) = 3%; pH 8.0; 70 °C; 20 min) and bromelain (B) (E/S = 5%; pH 6.7; 45 °C; 60 min), respectively. Tilapia fish skin gelatin (T), milkfish skin gelatin (M), halibut skin gelatin (H) and porcine skin gelatin (P) hydrolysates were prepared using Flavourzyme (F) (E/S = 3%, pH 7.0, 50 °C, 4 h) or Alcalase (A) (E/S = 3%, pH 8.0, 50 °C, 4 h), respectively. Tuna cooking juice (T) was hydrolyzed by protease XXIII (P) (E/S = 2.1%, pH 7.5, 37 °C, 60 min). After hydrolysis, the hydrolysate solutions were heated in boiling water for 15 min to inactivate the enzymes and were then cooled in water at room temperature for 20 min. Hydrolysates were adjusted to pH 7.0 with 2 M of NaOH and centrifuged (Centrifuge 05P-21, Hitachi Ltd., Katsuda, Japan) at 10,000× *g* and 4 °C for 10 min. The supernatant was lyophilized and stored at −20 °C.

### 2.4. Cell Culture

HSC-3 human oral squamous carcinoma and normal human oral keratinocytes (NHOK) were kind gifts from Dr. Chun-Yin Huang at China Medical University (Taichung, Taiwan). The HSC-3 cells were maintained in Dulbecco’s Modified Eagle’s Medium (DMEM)/F-12 and supplemented with 10% fetal bovine serum (FBS) and 1% antibiotic-penicillin/streptomycin. NHOK was cultured in keratinocyte serum-free medium (Invitrogen) supplemented with EGF in 0.1% BSA, 1% penicillin/streptomycin, L-Glutamine, and bovine pituitary extract. All cells were cultured with 5% CO_2_ at 37 °C. Cell viability was measured with the trypan blue method.

### 2.5. MTT Assay

To avoid a pH variation of the cell culture medium during sample solubilization, fish hydrolysate stock solution was prepared in 0.1 M PBS (pH 7.4). The cells were seeded in a 96-well microtiter plate (1 × 10^4^ cells/well) overnight, and then treated with various concentrations of hydrolysates and their ultrafiltration fractions. After incubating for 72 h, the effect of hydrolysates on cell growth was examined using the MTT (3-(4,5-dimethylthiazol-2-yl)2,5-diphenyl tetrazolium bromide) assay. About 20 μL of MTT solution (5 mg/mL, Sigma Chemical Co., Ltd., St. Louis, MI, USA) was added to each well and incubated at 37 °C for 4 h. The supernatant was aspirated and the MTT-formazan crystals formed by metabolically viable cells dissolved in 200 μL of isopropanol. Finally, the absorbance was read at 570 nm with a microplate reader. The hydrolysate concentration, which gives a 50% growth inhibition, is referred to as the IC_50_.

### 2.6. Western Blotting

Cells were washed with cold phosphate-buffered saline (PBS) and lysed with a solution containing 137 mM NaCl, 20 mM Tris (pH 7.4), 25 mM β-glycerophosphate, 0.1% Triton X-100, 2 mM EDTA, and 2 mM protease/phosphatase inhibitors. Protein concentration was determined using the Bradford protein-binding assay, and denatured proteins were separated in 8% to 12% SDS polyacrylamide gel electrophoresis and transferred onto PVDF membranes. Nonspecific binding was blocked with 5% milk in TBST buffer (20 mM Tris base, 140 mM NaCl, pH 7.6, 0.1% Tween-20) for 1 h, followed by incubation with primary antibodies at 4 °C overnight and secondary antibodies at room temperature for 1 h. Blots were visualized by ECL detection reagents.

### 2.7. Cell Cycle Analysis

For cell cycle analysis, cells were seeded at a density of 6 × 10^4^ cells/well in 6-well plates, cultured overnight, and then treated with various concentrations of PA hydrolysates. To analyze the cell cycle, after 72 h of treatment, the cells were harvested by trypsinization, washed in PBS, and fixed in 70% ice-cold ethanol. The cell pellets were resuspended in 500 μL of a solution containing 50 μg/mL of propidium iodide, 0.4 mg/mL of RNase A, 0.1% Triton-X-100 in PBS buffer, and then incubated at 37 °C for 30 min. The stained cells were subjected to DNA content/cell cycle analysis using an LSR flow cytometer (BD Bioscience Inc., Franklin Lakes, NJ, USA).

### 2.8. Apoptosis Analysis

For apoptosis, the Mitostatus Red and FITC Annexin V Apoptosis Detection Kit (BD Pharmingen^TM^, San Jose, CA, USA) were used to evaluate the mitochondrial membrane potential and assess annexin V-positive cells. To analyze the mitochondrial membrane potential, after 72 h of treatment, the cells were added to 2 mL of stain buffer with 200 nM BD Mitostatus Red and incubated for 30 min at 37 °C, protected from light. We removed the staining solution and washed cells in PBS buffer, and then cell pellets were resuspended in 500 μL of trypsin-EDTA buffer and incubated at 37 °C for 5 min. After incubation, 1.5 mL of stain buffer was added to each tube, and the cells were analyzed by using an LSR flow cytometer. To analyze the apoptosis, we washed the cells twice with cold PBS and then resuspended the cells in 1X annexin binding buffer. One hundred microliters of the solution were transferred to a 5 mL culture tube, and we added 5 μL of FITC Annexin V and 5 μL of PI. We gently vortexed the cells and incubated them for 15 min at room temperature in the dark. After incubation, 400 μL of 1X annexin binding buffer were added to each tube, and the cells were analyzed by using an LSR flow cytometer.

### 2.9. Statistical Analysis

Each data point represents the mean of three samples and was subjected to an analysis of variance (ANOVA) followed by Duncan’s multiple range test, and the significance level of *p* < 0.05 was employed.

## 3. Results and Discussion

### 3.1. Antiproliferative Activities of Protein Hydrolysates

According to the literature, protein hydrolysates showed their IC_50_ values between 0.5 and 426 mg/mL against various cancer cells [16]. We started with several food sources: hydrolysates from porcine skin gelatin (PA, PF), fish skin gelatin (TA, TF, HA, HF, MF), tuna cooking juice (TP), soy protein isolate (SB, ST), sodium caseinate (CB, CT), and wheat gluten (WB) at the concentrations of 2 and 5 mg/mL, and their antiproliferative effects on the oral cancer cell, HSC-3, after 24 or 48 h were shown, as seen in Figure 1A. All the protein hydrolysates possessed no significant antiproliferative activity at 24 h. At 48 h (Figure 1B); stronger antiproliferative activities (31.7~98.8%) were observed in MF, TP, SB, ST, CB, CT and WB hydrolysates (*p* < 0.05). The soy protein isolate hydrolysates (SB and ST) at 2 or 5 mg/mL showed significantly higher antiproliferative activities (96.2~98.8%, *p* < 0.01) than other hydrolysates. Therefore, the effects of SB and ST on the inhibition of HSC-3 cell growth were further investigated in detail.

From the results shown in Figure 1B, the hydrolysates from soy protein isolate (SB, ST), sodium caseinate (CB, CT) and wheat gluten (WB) showed higher antiproliferative activities; incidentally, all the protein hydrolysates from animal sources showed little effect on HSC-3 cell growth. It has been reported that anticancer peptides were rich in Cys, Gly, Ile, Lys, Tyr, Phe and Trp residues, while most antimicrobial peptides comprised Gly, Ala, Lys and Leu residues by an in silico analysis [17]. Therefore, we surveyed the relative abundance of all amino acids in these proteins. As shown in Appendix A, residues like Asp, Cys, Glu, His, Ile, Leu, Lys, Phe, Thr, Tyr and Val were found abundant in the hydrolysates from soy protein isolate, sodium caseinate and wheat gluten when compared to those from animal sources. These results suggest that Cys, Ile, Lys, Phe and Tyr in protein sequences may play a crucial role against cancer cells, which may reflect the novel in silico method having the chance as an excellent screening tool to predict the potential of a protein source as anticancer peptides.

### 3.2. Antiproliferative Activity of SB and ST

The effects of SB and ST at 0.5–2 mg/mL on the growth inhibition of HSC-3 for 48 and 72 h were shown in Figure 2. The multiple concentrations of SB and ST showed no effects on the growth of HSC-3 when compared to the control at 24 h (data not shown). At 48 h, the result showed significantly stronger growth inhibition (14.98–52.72%, *p* < 0.05) of SB and ST, particularly at higher concentrations of 1–2 mg/mL; and the IC_50_ values against HSC-3 were 4.01 and 1.69 mg/mL for SB and ST, respectively (Figure 2A and Table 1). However, the inhibition rates ranged from 35.45% to 76.39% in a concentration-dependent manner at 72 h; and the IC_50_ value against HSC-3 of SB and ST was 0.74 and 0.60 mg/mL, respectively (Figure 2B and Table 1). The cytotoxic effect of SB and ST at concentrations between 0.25 and 1 mg/mL were tested for 72 h by using a normal human oral keratinocytes (NHOK) cell line (Figure 2C). The results showed that ST promotes proliferative activities up to 140% and SB slightly inhibited cell growth to 82%; nevertheless, the SB and ST did not show an obviously cytotoxic effect on the cell viability of NHOK cells while those significantly inhibited the cell growth of HSC-3 in the same condition. This result is similar to a previous study that reported that the soybean protein hydrolysates (1 g/L) promoted the proliferation of human keratinocytes to 189%; however, higher concentrations (4 and 6 g/L) of the protein hydrolysates inhibited the cell growth because the resulting high concentration of amino acids or oligopeptides disturbs the nutrient balance [18]. Roe protein hydrolysates of giant grouper were investigated for their abilities to inhibit cell proliferation of oral cancer cells, and the ultrafiltration fraction (<5 kDa) showed the IC_50_ value against Ca9-22 cells was 0.85 mg/mL in terms of the ATP assay [15]. A peptide fraction (>10 kDa) of a soybean protein hydrolysate, after 6 days of germination, showed the IC_50_ values of 16.2, 14.3 and 15.2 mg/mL, respectively, against HeLa, CasKi and MDA-MB-231 cells [19]. SB and ST had relatively lower IC_50_ values against HSC-3 in the present study, as compared to those against various cancer cells. A previous study has revealed that Vismodegib, the first Hedgehog signaling pathway targeting agent approved by the US Food and Drug Administration (FDA) and European Medicines Agency (EMA), showed its IC_50_ value of 41.3 μg/mL against HSC-3, while the positive controls, Doxorubicin and 5-fluorouracil, as the inhibitors of Smoothened co-receptor, demonstrated the 50% cytotoxic activity at 0.19 and 2.18 μg/mL [20]. As compared to the agents approved for cancer treatment, protein hydrolysates show relatively lower cytotoxic activity on cancer cells and must be purified to obtain their fractions or bioactive peptides for therapeutic needs. A black soybean protein hydrolysate with the IC_50_ value of 1630 μg/mL against MCF-7 was purified by ultrafiltration, gel filtration and RP-HPLC in sequence, and its F2-c fraction potentially inhibited the cell proliferation with an IC_50_ of 68 μg/mL [21]. Therefore, the results indicated that the SB and ST would be candidates for the preparation of potentially antiproliferative peptides against HSC-3 cells.

### 3.3. Cell Cycle

As the regulation of the cell cycle is critical for the growth and development of cancer, we determined the effect of SB and ST on cell cycle progression. The results indicated that HSC-3 cells treated with SB (Figure 3A) or ST (Figure 3B) (concentrations ranging from 0, 0.25, 0.5 to 1.0 mg/mL) for 72 h showed significant cell accumulation in the S phase. The percentage of cells in the S phase increased from 13.68% to 23.84–28.55% after being treated with SB, while those increased from 11.23% to 44.11% at 0.25 mg/mL but decreased to 23.4% and 24.31% at 0.5 and 1.0 mg/mL of ST treatment. Interestingly, the cell population decrease in G0/G1 phase mirrored the increase in the S phase. These results reveal that the ST and SB treatments induced the cell cycle of HSC-3 arrest in the S phase. To examine the role of cell cycle-regulating proteins in the HSC-3 treated with SB or ST, proteins were extracted for Western blot analysis after the 72-h treatment. When compared with the control group, the expression of cyclin E and CDK2 were significantly decreased in a concentration-dependent manner for the SB treatment; whereas the expressions of p27 were decreased from 0.25 to 1 mg/mL, while those of p21 were unchanged (Figure 4). The ST treatment decreased the expression of cyclin E, cyclin A, p21 and p27 with the increases in ST concentrations, but that did not alter CDK2 expression. These results suggest that SB and ST may have inhibited cell proliferation by inducing cell cycle arrest in the S phase through decreases in cyclin E, cyclin A or CDK2 protein expression but independent of p21 and p27 protein expression. This phenomenon in the present study is different from those in other studies. Phenethyl isothiocyanate (PEITC) induced the downregulation of cyclins and CDKs and upregulation of the CDK inhibitor p21/p27 expression for causing G0/G1 phase arrest in HSC-3 cells [22]. A study demonstrated that orlistat increased the population of HSC-3 cells in the G2/M phase by deactivation of CDK1, and the reduction in viability of HSC-3 cells can be due to the inhibition of cyclin E expression [23]. Hinokitiol induced S-phase arrest in the cell cycle progression in HCT-116 cells and decreased the expression levels of cyclin A, cyclin E and CDK2 by increasing the expression of p21 [24]. These results in the present study suggest that SB and ST may have inhibited cell proliferation by inducing cell cycle arrest in the S phase through decreases in cyclin E, cyclin A or CDK2 protein expression but independent of p21 and p27 protein expression.

### 3.4. Apoptosis

Since cell apoptosis may be one of the consequences of cell cycle arrest, we examined whether SB or ST induced apoptosis in HSC-3 cells. We stained the cells with FITC Annexin V and Propidium Iodide (PI), and we conducted internucleosomal DNA fragmentation assays. In addition, we stained the cells with Mitostatus Red to detect mitochondrial membrane potential. As shown in Figure 5A,B, the mitochondria of HSC-3 cells become depolarized after 72 h incubation with SB or ST (0.25, 0.5 and 1.0 mg/mL) in a concentration-dependent manner. Since mitochondrial depolarization accompanies cytochrome C released during apoptosis, our results indicated that SB or ST might induce apoptosis of HSC-3 cells. The flow cytometry-based annexin V/PI patterns of SB- or ST-treated HSC-3 cells were performed (Figure 5C,D). The percentages of late apoptotic (Q2) cells increased only in those HSC-3 cells treated with higher concentrations of SB (1.0 mg/mL) to 31.5% and ST (0.5 or 1.0 mg/mL) to 60.4% and 64.4% (Figure 5E,F), respectively, while those of early apoptotic (Q4) cells remained almost unchanged. Moreover, an increase in the proportion of necrosis (Q1) cells was only observed at 1.0 mg/mL of SB treatment, but the number of necrosis cells was not changed after ST treatment at the concentrations tested. From the results, SB is speculated to induce apoptosis and necrosis while ST induced apoptosis of HSC-3 cells.

The expression of apoptosis-related proteins was investigated to analyze the underlying mechanisms by Western blot analysis. As shown in Figure 6, SB downregulated the expression of poly ADP-ribose polymerase (PARP) and upregulated the expression of Bax at the concentrations of 0.5 mg/mL; however, the other proteins did not show any significant changes. Treatment with ST resulted in a decrease in Bcl-2, PARP, caspase 3 and caspase 9 expression, and increases in p53 and cleaved caspase 3 expressions at the concentrations of 0.5 and 1 mg/mL. From the results, SB had no effect on the expression of apoptosis-related proteins, such as p53 and caspase 3. Therefore, we suggested SB maybe induce necrosis rather than apoptosis in HSC-3 cells. ST was supposed to induce apoptosis of HSC-3 cells by upregulating the modulator of apoptosis, p53, downregulating the apoptosis inhibitor, Bcl-2, and then activating the caspase-related proteins family and might be through the mitochondria-mediated pathway. This is in agreement with a previous study of berberine-induced apoptotic death of HSC-3 oral cancer cells via the mitochondrial pathway [25].

Food-borne bioactive peptides have been proven to possess several health benefits, such as antioxidative, antihypertensive, antihyperglycemic and anticancer, and they play an important role as suitable alternatives for consideration owing to their relative availability and affordability, along with their natural characteristics. Relative to antibodies or proteins, the advantages of peptides for therapeutic needs include small size, higher affinity, activity and improved penetration into cell membranes. Nonetheless, peptides have considerable limitations to cancer treatment due to their low bioavailability, short half-life in vivo and metabolic instability [16]. Hence, the practical approaches for developing oral peptide delivery systems with high bioavailability are critically important and need further studies to address.

## 4. Conclusions

Soy protein isolate hydrolysates, ST and SB, showed an antiproliferative effect, and induced cell cycle arrest in the S phase, and apoptosis and necrosis against HSC-3 cells. This study has clearly demonstrated that soy protein isolate hydrolysates have the potential to be a source of bioactive peptides with anti-oral cancer properties without affecting normal oral cells. Further studies are needed to purify the bioactive peptides, identify their sequences and elucidate their biological mechanisms of action in order to be potentially developed as functional foods.

## Figures and Tables

**Figure 1 molecules-27-02839-f001:**
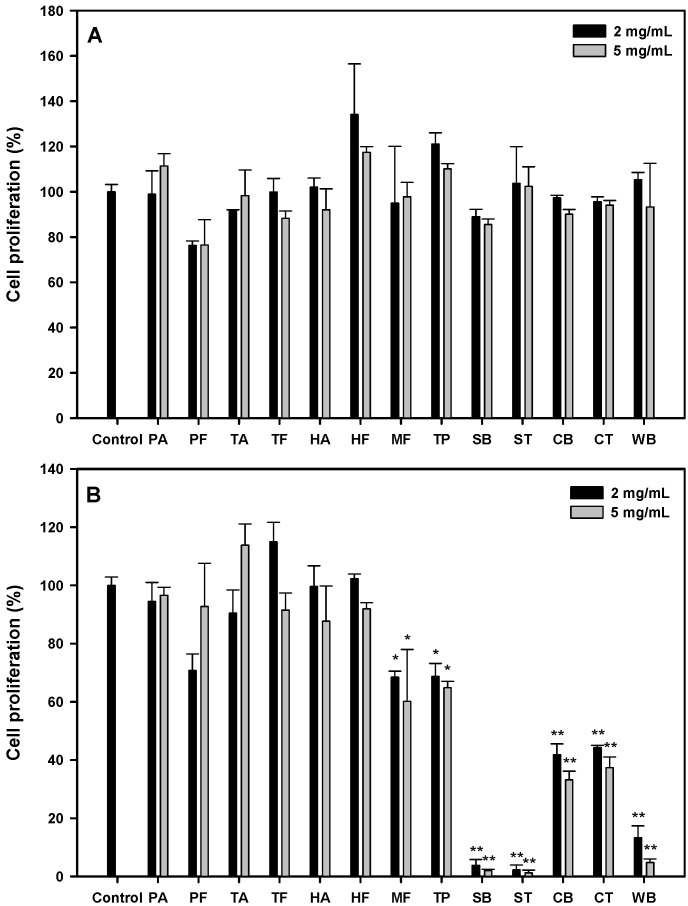
Cell proliferation of HSC-3 treated with various protein hydrolysates at the concentration of 2 and 5 mg/mL for 24 h (**A**) and 48 h (**B**). Data, mean ± SD (n = 3). * *p* < 0.05 and ** *p* < 0.01 against control. PA: Porcine skin gelatin/Alcalase; PF: Porcine skin gelatin/Flavourzyme; TA: Tilapia skin gelatin/Alcalase; TF: Tilapia skin gelatin/Flavourzyme; HA: Halibut skin gelatin/Alcalase; HF: Halibut skin gelatin/Flavourzyme; MF: Milkfish skin gelatin/Flavourzyme; TP: Tuna cooking juice/Protease XXIII; SB: Soy protein isolate/Bromelain; ST: Soy protein isolate/Thermolysin; CB: Sodium caseinate/Bromelain; CT: Sodium caseinate/Thermolysin; WB: Wheat gluten/Bromelain.

**Figure 2 molecules-27-02839-f002:**
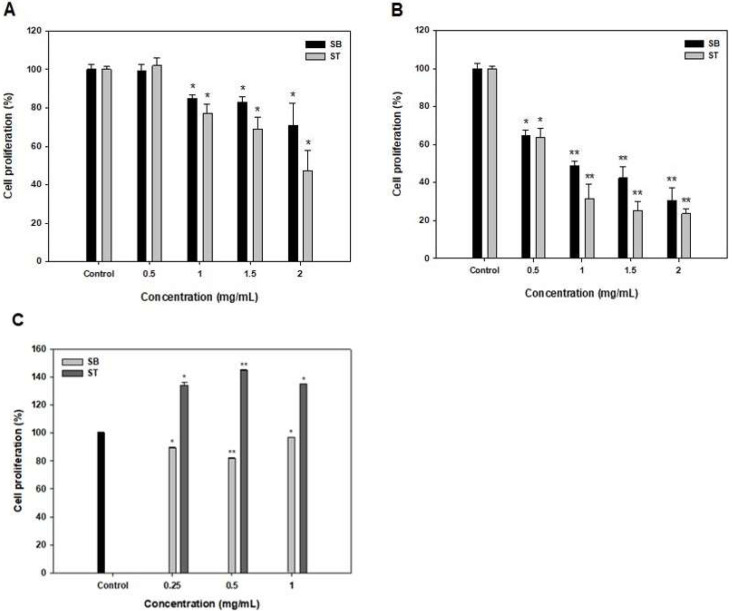
Effects of SB and ST treatments on cell proliferation of HSC-3 for 48 h (**A**), 72 h (**B**) and NHOK for 72 h (**C**) at various concentrations. Bars represent standard deviations from triplicate determinations. Data, mean ± SD (n = 3). * *p* < 0.05 and ** *p* < 0.01 against control.

**Figure 3 molecules-27-02839-f003:**
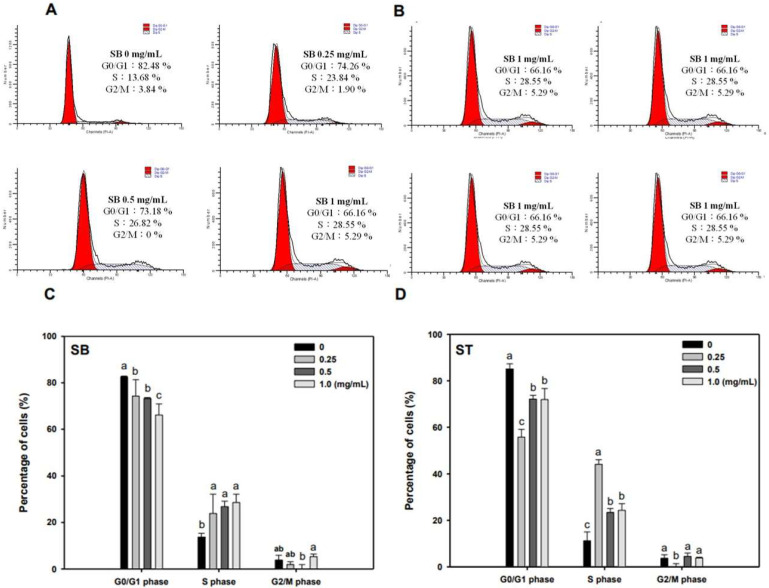
Effect of SB and ST on the cell cycle progression of HSC-3. (**A**,**B**) After the treatment of SB and ST at various concentrations for 72 h, HSC-3 cells were harvested, stained with propidium iodide, and analyzed by flow cytometry. Flow cytometric histograms are representative of three separate experiments. (**C**,**D**) Quantification of the percentage of HSC-3 cells treated by SB and ST in the cell cycle. Bars represent standard deviations from triplicate determinations. Different letters indicate significant differences (*p* < 0.05).

**Figure 4 molecules-27-02839-f004:**
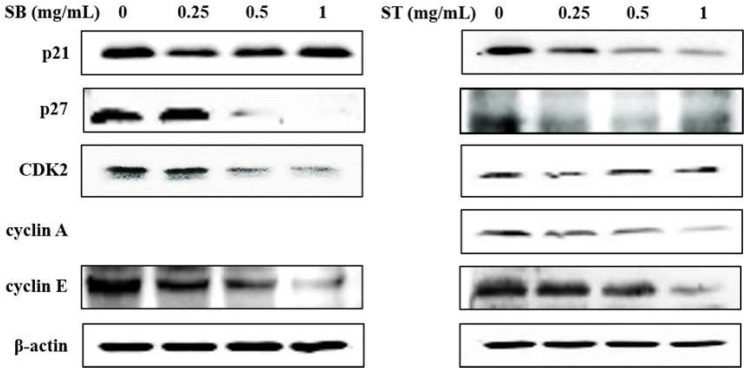
Effect of SB and ST on apoptosis-related protein levels in HSC-3 cells investigated by Western blot analysis. Cells were treated with 0.25, 0.5 and 1 mg/mL of SB and ST for 72 h. β-actin was used as a loading control. Figures showed the representative blots from one of three experiments that gave similar results.

**Figure 5 molecules-27-02839-f005:**
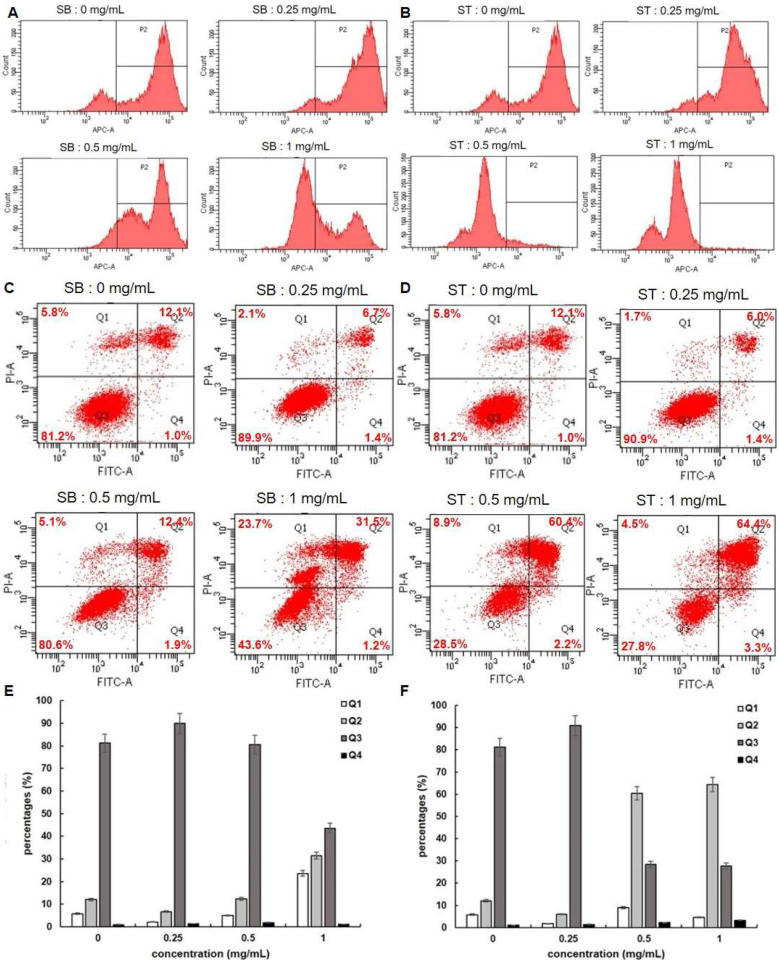
Apoptosis assessment of HSC-3 cells treated with indicated concentrations of SB and ST. (**A**,**B**) Detecting mitochondrial depolarization with Mitostatus reagents. (**C**,**D**) Flow cytometric analysis of PS externalization (annexin V binding) and cell membrane integrity (PI staining). Cells were treated with 0, 0.25, 0.5 and 1 mg/mL of SB and ST for 72 h. The dual parameter dot plots combining annexin V-FITC and PI fluorescence show the vial cell population in the lower left quadrant (Q3), the early apoptotic cells in the lower right quadrant (Q4), the late apoptotic cells in the upper right quadrant (Q2), and the necrosis cells in the upper left quadrant (Q1). (**E**,**F**) Percentages of Q1, Q2, Q3 and Q4 cells. Bars represent standard deviations from triplicate determinations.

**Figure 6 molecules-27-02839-f006:**
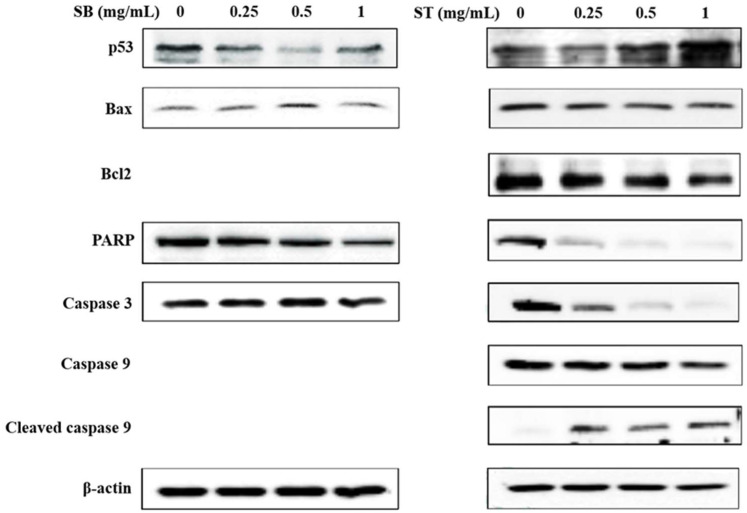
Effect of SB and ST on apoptosis-related protein levels in HSC-3 cells investigated by Western blot analysis. Cells were treated with 0.25, 0.5 and 1 mg/mL SB and ST for 72 h. β-Actin was used as a loading control. Figures show the representative blots from one of three experiments that gave similar results.

**Table 1 molecules-27-02839-t001:** IC_50_ values of SB and ST on HSC-3 at 48 and 72 h.

Treated Time (h)	IC_50_ (mg/mL)	
	SB	ST
48	4.01	1.69
72	0.74	0.60

## Data Availability

The data presented in this study are available on request from the corresponding author.

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
