# Peer review of "The Hydrolytic Peptides of Soybean Protein Induce Cell Cycle Arrest and Apoptosis on Human Oral Cancer Cell Line HSC-3"

_molecules, 2022, doi:10.3390/molecules27092839_

Round 1

Reviewer 1 Report

This manuscript reports on the cytotoxic activity of protein hydrolysates of several food and the apoptosis-inducing activity of the hydrolytic peptides of soy bean proteins in HSC-3 oral cell lines. Standard procedures were used but the experimental design is rather weak. The cytotoxic activity of the peptides at such a high concentration is of little biological importance.

  1. The hydrolysates showed cytotoxic activity at very high concentrations up to 1.0 mg/ml after 72 h. This is considered very weak. Did the authors compared their values with those in the literature? What is the rationale to investigate the mechanism of action further if the IC50 is very high to begin with?
  2. Figure 2A and B: Are both graphs (left: viability, right: antiprolferation) derived from the same set of data? Please do not duplicate the same data in two different graphs.
  3. The choice of terminology is important. The MTT assay shows cell viability and should not be inferred as "antiproliferative activity" of the tested substance.
  4. Figure S2 should be included in the manuscript.
  5. Figure 4: the protein bands for p27 (ST) is not clear
  6. Figure 5: treatment with the extracts did not cause an increased in early apoptotic cells but a direct increase in late apoptotic cells, but the authors postulated that ST induced apoptosis while necrosis for SB.  The observation in the annexin V assay doesn't seem to support the hypothesis. Please explain.
  7. The experimental design is weak as no appropriate positive controls were used in any of the assays, hence, lowering the overall impact of this study.
  8. There is little to no attempt to compare their data with those reported in the literature.
  9. If the peptides exhibit activity at such a high concentration, then I am unsure if these can still be considered "good dietary supplement".

Author Response

Comments and Suggestions for Authors

This manuscript reports on the cytotoxic activity of protein hydrolysates of several food and the apoptosis-inducing activity of the hydrolytic peptides of soy bean proteins in HSC-3 oral cell lines. Standard procedures were used but the experimental design is rather weak. The cytotoxic activity of the peptides at such a high concentration is of little biological importance.

Response: Thank you so much for your comments and suggestions on our manuscript. And revised portions are marked in red in the revised manuscript so that they may be easily identified.

1. The hydrolysates showed cytotoxic activity at very high concentrations up to 1.0 mg/ml after 72 h. This is considered very weak. Did the authors compared their values with those in the literature? What is the rationale to investigate the mechanism of action further if the IC50 is very high to begin with?

Response: In the beginning of this research, we screened the various protein hydrolysates which may have the potential to inhibit the growth of HSC-3 cells, and the protein hydrolysates with higher antiproliferative activities were selected for further investigation. According to the literature, protein hydrolysates showed their IC50 values between 0.5 and 426 mg/mL against various cancer cells (Sharma et al. 2021), such as roe protein hydrolysates had the IC50 value of 0.85 mg/mL for oral cancer Ca9-22 cells but IC50 value was undetectable (>2.5 mg/mL) for CaL 27 cells (Yang et al., 2016). Black soybean protein hydrolysate showed anticancer activity with the IC50 values of 0.564, 1.63 and 1.84 mg/mL on HepG2, MCF-7 and Hela cancer cells (Chen et al., 2019). Therefore, we used the concentrations of 2 and 5 mg/mL of protein hydrolysates to evaluate their antiproliferative activities. SB and ST which showed the IC50 values of 0.74 and 0.60 mg/mL, respectively, against HSC-3 after 72-h incubation have great potential as the precursor of anticancer peptides as compared to the other protein hydrolysates reported in literature. The information about the reason we selected the concentrations for determination of cytotoxic activity on HSC-3 has been added in the text of the manuscript.

2. Figure 2A and B: Are both graphs (left: viability, right: antiprolferation) derived from the same set of data? Please do not duplicate the same data in two different graphs.

Response: We’ve deleted the right graphs in Figure 2A and B and added Table 1 for presenting the IC50 values of SB and ST against HSC-3.

3. The choice of terminology is important. The MTT assay shows cell viability and should not be inferred as "antiproliferative activity" of the tested substance.

Response: We’ve changed the wording of “antiproliferative activity” to “growth inhibition” based on the results of the MTT assay.

4. Figure S2 should be included in the manuscript.

Response: We’ve removed Figure S2, and the results of cell viability of NHOK treated with SB and ST were inserted in Figure 2C.

5. Figure 4: the protein bands for p27 (ST) is not clear

Response: We’ve changed the unclear photo by using the original file of p27 photo.

6. Figure 5: treatment with the extracts did not cause an increased in early apoptotic cells but a direct increase in late apoptotic cells, but the authors postulated that ST induced apoptosis while necrosis for SB.  The observation in the annexin V assay doesn't seem to support the hypothesis. Please explain.

Response: The percentages of late apoptotic (Q2) cells increased only in those HSC-3 cells treated with higher concentrations of SB (1.0 mg/mL) and ST (0.5 or 1.0 mg/mL) while those of early apoptotic (Q4) cells almost unchanged. From the results, SB and ST induced apoptosis or necrosis of HSC-3 cells needed further analyses of the expression of apoptosis-related proteins. Based on the results, SB was observed to show no effect on the expression of apoptosis-related proteins, while ST upregulated p53, downregulated Bcl2 and then activating caspase-related proteins family. Therefore, we suggested that SB induced necrosis rather than apoptosis while apoptosis for ST.

7. The experimental design is weak as no appropriate positive controls were used in any of the assays, hence, lowering the overall impact of this study.

Response: Actually, there is no appropriate positive controls for HSC-3. Although a previous study has used a positive control “EGF” (human recombinant epidermal growth factor) for oral tongue squamous cell carcinoma cell lines (HSC-3, SCC-25 and SAS), EGF did not show any effect on cell proliferation of HSC-3 after 24, 48 and 72 h-treatment (Vierthaler et al., 2020). Also, recent studies focused on the inhibitory effects of peptides on HSC-3 growth did not use positive controls (Chen et al., 2020; Yang et al., 2021). Therefore, no positive controls were used in the present study because of being lack of an appropriate positive control for HSC-3 cells.

8. There is little to no attempt to compare their data with those reported in the literature.

Response: We have added more discussion about the mechanism and the comparison between the data in the present study and that in the literature.

9. If the peptides exhibit activity at such a high concentration, then I am unsure if these can still be considered "good dietary supplement".

Response: We’ve rephrased the sentence about the “good dietary supplement”, and the conclusion we stated that soy protein hydrolysates have the potential to be a good source of bioactive peptides with the anti-oral cancer property with affecting normal oral cells. Further studies are needed to purify the bioactive peptides, identify their sequences and elucidate their biological mechanisms of action in order to be potentially developed as functional foods.

Reviewer 2 Report

I find the subject of the research in the manuscript interesting, but there are some inconsistencies and the presentation of the results could be improved.

I have several comments:

  • The general aspect of the paper – there are some typing and language mistakes and some errors related to the form/template (Materials and method - Statistical analysis – page 4 – should be 3.0. and the lining is wrong; Title for Figure 2);
  • I understand that the focus of the study is on the peptides obtained from soybean, but I would consider mentioning the other protein hydrolysates investigated in the abstract. It is just a suggestion; the authors could reflect on it.
  • Materials and methods – 2.8. – specify the type of flow cytometer;
  • Results and discussion – 3.1. – amino acids found in the protein hydrolysates – Figure S1 – the cited references are very old; more recent references should be found;
  • Results and discussion – 3.3. – Western blotting – Figure 4 – why is the cyclin A section missing for SB treatment? Similar question for Figure 6.

The authors mentioned: When compared with the control group, the expression of cyclin E, cyclin A and CDK2 were significantly decreased in a concentration-dependent manner for the SB or ST treatment; whereas the expressions of p21 and p27 were decreased at 0.25 to 1 mg/mL. The CDK2 expression does not seem significantly decreased for ST treatment; the same for p21 for SB treatment. The authors might consider explaining or rephrasing in order to make the statement more clear.

  • 4. Apoptosis – flow cytometry Figure 5 C/D – add the percentage of cells located in the four quadrants for a clearer image of late/early apoptosis;
  • I could not find in the results section the data about the identification of amino acid sequence by MALDI-TOF:
  • In my opinion, the paper would benefit greatly if the discussion section would be improved.
  • The number of references is reduced and they are not very recent. The references are not written according to the template of the journal.

Author Response

I find the subject of the research in the manuscript interesting, but there are some inconsistencies and the presentation of the results could be improved.

Response: Thank you so much for your comments and suggestions on our manuscript. And revised portions are marked in red in the revised manuscript so that they may be easily identified.

I have several comments:

  • The general aspect of the paper – there are some typing and language mistakes and some errors related to the form/template (Materials and method - Statistical analysis – page 4 – should be 3.0. and the lining is wrong; Title for Figure 2);

Response: Some typing and languages mistakes were revised and marked in red in the manuscript. Materials and method – Statistical analysis is revised to section 2.9. on page 4. The title for figure 2 was revised, which contains effects of SB and ST treatments on cell proliferation of HSC-3 for 48 h (A), 72 h (B) at various concentrations. We also make the data of figure S2 change into figure 2 (C), which mean Effects of SB and ST treatments on NHOK for 72 h at various concentrations. We considered these changes can make the expression clearer.

  • I understand that the focus of the study is on the peptides obtained from soybean, but I would consider mentioning the other protein hydrolysates investigated in the abstract. It is just a suggestion; the authors could reflect on it.

Response: We revised the abstract to make it more clear to understand the other protein hydrolysates and why we focus on the peptides obtained from soybean in this study. Please check the corrected abstract in the manuscript.

  • Materials and methods – 2.8. – specify the type of flow cytometer;

Response: We mentioned the type of flow cytometer in Materials and methods – section 2.7. Cell cycle analysis. The type of flow cytometer that we used is an LSR flow cytometer (BD Bioscience Inc., Franklin Lakes, NJ, USA). Therefore, we didn’t repeat the description in Materials and methods – section 2.8. Apoptosis analysis.

  • Results and discussion – 3.1. – amino acids found in the protein hydrolysates – Figure S1 – the cited references are very old; more recent references should be found;

Response: We searched the cited references and changed the references to other recent sources: the amino acid composition data was cited from Hafidz et al. (2011) (porcine skin gelatin), Wang et al. (2015) (fish skin gelatin), Rombouts et al. (2009) (wheat gluten), Cervantes-Pahm and Stein (2010) (soy protein isolate), Mackle et al. (1999) (sodium caseinate).

  • Results and discussion – 3.3. – Western blotting – Figure 4 – why is the cyclin A section missing for SB treatment? Similar question for Figure 6.

Response: The Western blotting for SB in Figure 4 and Figure 6 is different from that for ST because SB decreased the check point cyclin E at late G1 phase and CDK2 expression. Also, the percentage of cells in the G2 phase was below 5.29%. Therefore, cyclin A, the check point at S phase was not determined in Figure 4.

          Similar reason for Figure 6, SB had no effect on the expression of apoptosis-related proteins, such as p53, caspase 3 and Bax, therefore, we postulated that SB induced necrosis of HSC-3 cells by the accumulation of late apoptotic cells but not triggered apoptosis. The other apoptosis-related proteins for SB were not determined because the present results could explain the mechanism of action of SB.

The authors mentioned: When compared with the control group, the expression of cyclin E, cyclin A and CDK2 were significantly decreased in a concentration-dependent manner for the SB or ST treatment; whereas the expressions of p21 and p27 were decreased at 0.25 to 1 mg/mL. The CDK2 expression does not seem significantly decreased for ST treatment; the same for p21 for SB treatment. The authors might consider explaining or rephrasing in order to make the statement more clear.

Response: We’ve rephrased the whole paragraph and added explanations and discussion in order to make the statement clearer.

  • 4. Apoptosis – flow cytometry Figure 5 C/D – add the percentage of cells located in the four quadrants for a clearer image of late/early apoptosis.

Response: We’ve added the percentage of cells located in the four quadrants for a clearer of late/early apotosis. Please check the corrected Figure 5 C/D in the manuscript.

  • I could not find in the results section the data about the identification of amino acid sequence by MALDI-TOF:

Response: The description of identification amino acid sequence by MALDI-TOF in the original manuscript section 2.9. was implanted. We already removed the wrong description and section 2.9. will be Statistical analysis. Please check the corrected section in the manuscript.

  • In my opinion, the paper would benefit greatly if the discussion section would be improved.

Response: Thanks for your suggestion. We have added more discussion about the mechanism and the comparison between the data in the present study and that in the literature.

  • The number of references is reduced and they are not very recent. The references are not written according to the template of the journal.

Response: We have changed the references to the template of the journal. Please check the corrected references in the manuscript.

Round 2

Reviewer 1 Report

The authors have addressed most of the comments raised but 

  1. Please include the additional details in the revised manuscript especially #6.
  2. Positive controls mentioned in the last report referred to a known cytotoxic drug. The authors can compare the IC50 of the protein hydrolysates with the drug, and arrive at a better conclusion regarding the potency of their samples.
  3. In the discussion section, the authors should critically assess the suitability of protein hydrolysates since there is a possibility that these might be further digested by human digestive enzymes.

Reviewer 2 Report

I am satisfied with the revised form of the paper and I have no further comments.

Author Response

The response to the reviewer’s comments are listed below point by point.

Reviewer 2

I am satisfied with the revised form of the paper and I have no further comments.

Response: Thank you for your professional suggestions, and we are glad to hear you satisfied at our study.